# Exploring Sparse Adapters for Scalable Merging of Parameter Efficient Experts

**Samin Yeasar Arnob**
McGill University, Mila, Microsoft
samin.arnob@mail.mcgill.ca

**Zhan Su**
Université de Montréal

**Minseon Kim**
Microsoft

**Oleksiy Ostapenko**  **Riyasat Ohib**
ServiceNow        Georgia Institute of Technology

**Esra'a Saleh**
Université de Montréal, Mila

**Doina Precup**        **Lucas Page-Caccia**        **Alessandro Sordoni**
McGill University, Mila     Microsoft              Microsoft

## Abstract

Model merging aims to integrate knowledge from multiple task experts into a single, unified multi-task model. Parameter-efficient adaptation, namely LoRA, have become the de-facto approach to obtain memory-friendly task experts. In this paper, we study the properties of sparse adapters, which train only a subset of weights in the base neural network, as potential adaptation recipe for downstream model merging. First, we propose a simple method for training highly effective sparse adapters, which surprisingly outperforms both LoRA and full fine-tuning in our setting. Next, we investigate the merging properties of these sparse adapters, merging up to 20 natural language processing task adapters. Our findings demonstrate that sparse adapters yield superior in-distribution performance post-merging compared to LoRA or full model merging. Achieving strong held-out performance remains a challenge for all methods considered.

## 1 Introduction

Multitask training, e.g. (Raffel et al., 2019), is an effective method to improve the performance of large language models (LLMs) across different tasks. However, for multitask training, all task-specific datasets need to be available simultaneously requiring data sharing during training. Model merging has emerged as an efficient alternative to building multi-task models (Wortsman et al., 2022), which allows tasks to be trained separately and then combined at the end of the training process, thus ensuring privacy of data and savings at training time. Recent work shows that model averaging can improve out-of-distribution performance over multitask training (Yadav et al., 2024b; Ostapenko et al., 2024). However, merging models do not achieve the same performance as true multitask training due to weight interference, and requires careful weight manipulation (Ilharco et al., 2023; Yadav et al., 2023; Akiba et al., 2024; White, 2016; Davar, 2024) during the merging process to resolve conflicts.

Merging has recently been the focus of modular architectures that re-use parameter-efficient experts such as LoRA (Hu et al., 2021b) readily available on platforms such as Huggingface Hub (Huang et al., 2024; Ostapenko et al., 2024; Muqeeth et al., 2024). Recent evidence suggests that carefully composing LoRA (Hu et al., 2021a) modules can even outperform multi-task training on some tasks (Prabhakar et al., 2024). The widespread adoption of LoRA is due to the fact that it reduces the number of task-specific trainable parameters via low-rank decomposition while still maintaining good task performance. However, merging task-specific LoRA experts result in significant parameter interference given that all parameters of a given layer are modified for each task. In contrast, sparse fine-tuning methods — an alternative parameter-efficient fine-tuning methods that train a smaller

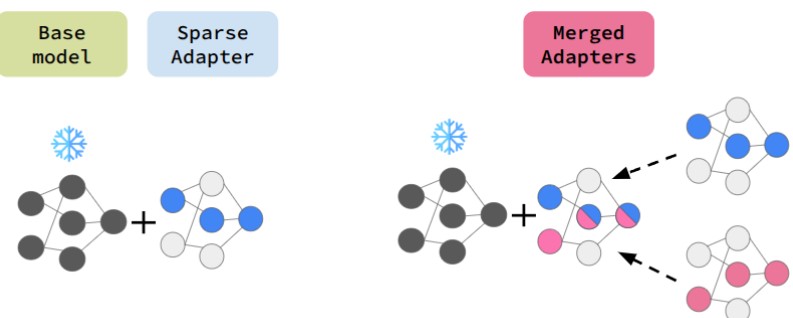

Figure 1: *Left:* The base model's weights remain frozen during training, with only the sparse adapter modules updated; during each forward pass, the adapter's outputs are combined with the fixed base weights. *Right:* Multiple sparse adapters—each trained separately on different tasks—are merged at inference time with the frozen base model weights.

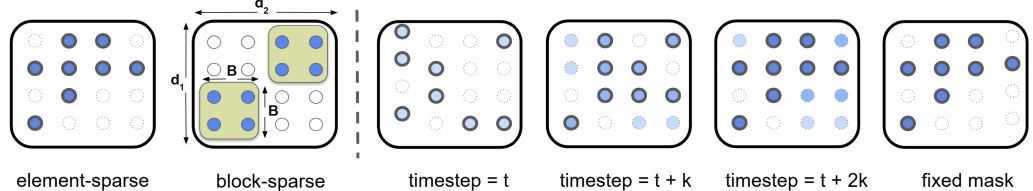

Figure 2: *Left:* Visualization of a weight matrix with sparsity ratio 0.5. Trainable parameters are highlighted in blue. Element-sparse and block-sparse with block size $B = 2$ and $N_B = 2$. *Right:* A visual representation of sparse-adapter reconfiguration during fine-tuning. Color transitions indicate updates to the weight-space via masking. Subsets of trainable parameters are updated periodically. Once the mask is fixed after a fixed number of steps, the weight-space remains constant, discarding unmasked weights.

sub-network within the base model for each task, have been proposed (He et al., 2022; Ansell et al., 2022; 2024; Panda et al., 2024) and have shown some promising results in composability (Ansell et al., 2022), albeit in a constrained setting of merging a language expert with a task expert for multi-lingual tasks. In addition, Su et al. (2024) have shown that sparse adapters are more suitable for concurrent serving scenarios than LoRA further motivating their use for modular scenarios.

In this paper, we study the properties of sparse adapters as potential building blocks for modular architectures. We begin by introducing a simple yet straightforward method for training sparse adapters, which simplifies prior approaches (Ansell et al., 2024; Panda et al., 2024) while delivering superior performance compared to LoRA and full fine-tuning. We leverage *connection sensitivity* (Mozer & Smolensky, 1988; Lee et al., 2018) to identify an important subset of parameters that are essential for the task. While structural sparse adapters are well-suited for concurrent serving (Su et al., 2024), we hypothesize that block-sparsity can also facilitate efficient merging, analogous to how block matrix-matrix multiplication accelerates expert routing in Gale et al. (2023). Motivated by this, we propose a simple approach to learn *block-sparse* adapters and find that they perform comparably to unstructured sparse adapters. We then explore the merging behavior of sparse adapters by conducting experiments across a set of 20 FLAN (Longpre et al., 2023) tasks—a significant expansion over previous work, which typically focuses on just a few tasks (Ansell et al., 2022; Panda et al., 2024). We compare the performance of sparse adapters with LoRA and full fine-tuning. We evaluate performance on both the test sets of the 20 in-distribution tasks (held-in) and a set of unseen 10 tasks (held-out). Additionally, we benchmark recent merging methods, including Task Arithmetic (Ilharco et al., 2023), Ties (Yadav et al., 2023) and Breadcrumbs (Davar, 2024).

Our results show that sparse adapters outperform both LoRA and full fine-tuning in a single fine-tuning experiment across 20 tasks. Moreover, merging sparse adapters retains strong held-in performance while maintaining competitive held-out results. Unlike full-finetuning merging methods, which degrade in performance when scaled to 20 experts, sparse-adapters prove to be more effective. Overall, our work demonstrates that sparse adapters offer a scalable and efficient approach to merging modular architectures for multi-task learning, especially when extending beyond the typical two-task setting explored in previous research. While we show improved generalization for model-merging, a notable gap remains compared to multitask performance.

## 2 Related Work

**Parameter-Efficient Finetuning (PEFT)** enable the efficient adaptation of LLMs through updating only a small subset of parameters (Han et al.). PEFT approaches directly update the pre-trained weights in a parameter-efficient manner (Hu et al., 2021a; Zhang et al., 2023; Hayou et al.; Liu et al., 2024; Dettmers et al., 2024). The most prominent method is Low-Rank Adaptation (LoRA) (Hu et al., 2021b), which parameterizes incremental weight updates $\Delta$ is the product of two low-rank matrices. LoRA achieves performance comparable to or even surpassing that of full fine-tuning. More recently (Hu et al., 2025) introduced a follow-up LoRS method, that achieves better computing and memory efficiency for fine-tuning sparse LLMs.

Parameter-sparse training has become increasingly popular in deep learning for achieving results similar to parameter-dense training (Frankle & Carbin, 2019; Lee et al., 2018; Wang et al., 2020; Evci et al., 2021; Arnob et al., 2021; 2024). Recent research on training sparse networks for LLMs has mainly concentrated on single-task training and merging with a limited number of tasks. For instance, He et al. (2022) evaluates the performance of various sparse training techniques for LLMs in a single-task context. Ansell et al. (2022); Panda et al. (2024) explore iterative magnitude pruning (Frankle & Carbin, 2019), and Ansell et al. (2024) introduces an evolution-based sparse training approach that adopts the prune-and-grow method described by Evci et al. (2021). Moreover, Ansell et al. (2022); Panda et al. (2024); Serra et al. (2018) show that sparsity can help prevent catastrophic forgetting. Structural sparse adapters impose specific structure on the non-zero elements of the adapters, e.g. OFT (Qiu et al., 2023) enforces a block-diagonal structure on it's orthogonal sparse adapter, while BOFT (Liu et al., 2023) uses butterfly factorization in order to improve the flexibility of OFT while keeping the block-spare and orthogonal structure of the resulting adapter, and S²FT (Yang et al., 2024b) leverages structured sparsity patterns to enable efficient and scalable fine-tuning across diverse LLM architectures. In this work, we scale sparse adapter merging to a broader multi-task setting involving 20 tasks. We asynchronously fine-tune sparse experts on the FLAN dataset (Longpre et al., 2023), and evaluate their performance under various model merging strategies. Motivated by potential efficiency gains in batched for batched inference scenarios, we investigate the performance of block-sparse adapters (Su et al., 2024), yet we do not enforce the orthogonality constrain as in OFT or BOFT leaving this direction for future work.

**Expert Merging** There is growing interest in aggregating adapters from diverse domains through model merging techniques (Yadav et al., 2024a). The simplest form of merging involves averaging the weights of different experts. Expanding on weight averaging, Task Arithmetic (Ilharco et al., 2023) involving the creation and combination of task vectors facilitated multi-task learning. Beyond simple averaging, Yadav et al. (2023) propose TIES, and Akiba et al. (2024) introduce DARE, both of which reset redundant parameters, resolve sign conflicts, and selectively merge parameters that demonstrate sign consistency. Similarly, Davar (2024) propose Breadcrumbs, a method that eliminates weight outliers and filters out negligible perturbations. Some methods like Fisher Merging (Matena & Raffel, 2022a) and RegMean (Jin et al., 2023b) need training data-based pre-computations to measure individual parameter importance but these are highly memory and data intensive. All of these merging methods require tuning merging hyper-parameters and carefully updating weights to avoid conflicts during model merging. We demonstrate that sparse adapters can be merged using simple weight averaging, yielding the best performance on held-in

---

**Algorithm 1** Sparse Adapter Training

---

**Init:** # Base params, Trainable params, Sparse Mask
▷ $W \in \mathbb{R}^{d_1 \times d_2}$, $\hat{W} = \mathbf{0}^{d_1 \times d_2}$, $M = \mathbf{1}^{d_1 \times d_2}$
▷ Training dataset: $D_{\mathcal{T}}$, optimizer($\hat{W}$)
**Training loop:**
**for** epoch = 1 to 5 **do**
   **for** step = 1 to N **do**
      batch $\sim \mathcal{D}_{\mathcal{T}}$
      loss = model(batch, $W + \hat{W} \cdot M$)
      loss.backward()
      optimizer.step()
      **if** epoch == 1 and step % 100 == 0 **then**
         loss = model(batch, $W + \hat{W}$)
         loss.backward()
         $M = \text{Top}_K(\hat{W} \cdot \hat{W}.\text{grad})$ # Eq. 2
      **end if**
   **end for**
**end for**

---

datasets while maintaining competitive generalization on held-out tasks. While Holtermann et al. (2024) focuses on composing knowledge modules for zero-shot generalization through optimal domain composition, our work focuses on studying the specific properties of sparsity in the context of model merging.

## 3 Training and Merging Sparse Adapters

We learn a sparse adapter for a task, where the task-dependent shift to the base model weights $\Delta W$ is sparse, $\Delta W = \hat{W} \cdot M$, where $M$ has entries in $\{0, 1\}$ and $\hat{W}$ is a dense weight. We initialize the trainable parameters $\hat{W}$ as zeros and learn the task specific mask $M$ using a parameter selection criteria that we present next. We propose two Sparse Adapters: **(a)** Element-sparse, where each parameter in a linear layer is scored individually, and **(b)** Block-sparse adapter, where we consider non-overlapping blocks of size $B$ within a linear layer. To ensure scalability when training large language models, we focus on training only the Query-Key-Value ($QKV$) layers. We justify our choice of layer selection for sparse fine-tuning through hyperparameter tuning, which is detailed in Fig. 7 in the Appendix A.

**Parameter Selection Criteria** Our parameter selection criterion is driven by the saliency-based score proposed by Mozer & Smolensky (1988), which has since been demonstrated to be effective in deep learning (Lee et al., 2018; Arnob et al., 2021; 2024) as a method for ranking parameters based on their importance at initialization.

The saliency-based score is inspired by "connection sensitivity" (SNIP) (Lee et al., 2018), which measures the influence of each parameter in the neural network for a given task:

$$\text{SNIP}(w_q) = \left| w_q \frac{\partial \mathcal{L}}{\partial w_q} \right|, \tag{1}$$

where $\delta_q$ is a vector whose $q$-th element equals $w_q$ and all other elements are 0. In practice, this is computed with a single forward-backward pass. Instead of computing the above, we find that it is empirically more effective to remove the absolute value and score parameters with the maximum connection sensitivity (MCS):

$$\text{MCS}(w_q) = w_q \frac{\partial \mathcal{L}}{\partial w_q}, \tag{2}$$

By not taking an absolute value on the score, we are ensuring that the learned weights $w_q$ and the current gradient estimate $\frac{\partial \mathcal{L}}{\partial w_q}$ have the same sign. Intuitively, this promotes selection for weights whose gradients consistently point in the same direction.

The mask $M$ retains parameters based on the saliency scores above a given threshold. We use keep-ratio ($kr$) to represent model sparsity, where $kr$ denotes the proportion of parameters that are trainable. For instance, a 95% sparse model corresponds to a $kr$ value of 0.05, meaning only 5% of the parameters are kept trainable. Accordingly, we set $k$ in Equation 2 as $kr \times (d_1 \times d_2)$.

**Block Sparsity** We present a simple strategy to identify block sparsity. Instead of calculating importance per-weight, we identifies and retains important regions within a linear layer. Given a keep-ratio value $kr$ and a linear layer of size $d_1 \times d_2$, the number of blocks is calculated as:

$$N_B = kr * \frac{(d_1 \times d_2)}{B \times B}, \tag{3}$$

where $B$ denotes the block size. We compute the importance score $\mathcal{S}$ for each non-overlapping block (by summing weight importance within blocks) and mask the parameters within the top $N_B$ blocks. We refer to block-wise sparse training as Block-Sparse. Fig. 2 illustrates an example of a Block-Sparse weight matrix. Block-wise sparsity is traditionally used for hardware acceleration. In this work, our focus is on exploring how far block-sparse methods can be improved in the context of model finetuning and model merging.

**Algorithm** See Algorithm 1. We initialize $\hat{W} = 0$ and $M = 1$. During the first epoch of fine-tuning, both $\hat{W}$ and $M$ are updated. The mask $M$ is recalculated using Eq. (2) every 100 gradient steps: we compute Eq. (2) after the backward pass to derive the importance score for each weight. We then select the top-$k$ parameters based on these scores to construct an updated mask $M$. Although the full parameter matrix is stored during this phase to enable mask selection, only a fixed sparse subset (e.g., 5%) is active and updated. Once the first epoch ends, the mask is fixed, the remaining weights are discarded, and only the active subset is trained for the rest of fine-tuning. This design means the reported percentage of trainable parameters accurately reflects the true optimization budget, while producing lightweight adapters that are convenient for sharing and merging. The overall algorithm is in Alg. 1 and a visual illustration of trainable weight-space update through masking in Fig. 2.

The algorithm requires periodic updates of the mask during the first epoch of fine-tuning. After the first epoch, the mask is kept fixed for the remainder of the fine-tuning. While Lee et al. (2018) proposes the saliency criteria in Eq. 1 for single-shot pruning (single forward and backward step for pruning), we find a significant improvement due to the periodic mask update. Performance comparison is shown in Fig. 8.

**Comparison with Prior Sparse Fine-tuning** Previous sparse fine-tuning approaches (Ansell et al., 2022; Panda et al., 2024) require a full fine-tuning for at least one epoch to identify the sparse mask $M$. In contrast, as shown in Algorithm 1, we can maintain sparsity from the start without needing a full fine-tuning stage. Ansell et al. (2024) presents a memory-efficient solution by preserving sparsity throughout training using a grow-and-drop method. Since we only sparse fine-tune the $QKV$ layers, the memory requirements do not increase significantly. We discuss the memory utilization in the Appendix C.

**Merging Sparse Adapters** For each task $\mathcal{T}_i$, $i = \{1, \ldots, N\}$ we have a sparse task-specific shift $\Delta W_i = \hat{W}_i \cdot M_i$. To perform the adapter merging across tasks, we need to properly account for the fact that some individual weights might be trained by two or more tasks. In particular, if a weight element is shared across $k$ different task-specific masks, we need to average the updates for that weight element by dividing the sum by $k$. We compute a weight overlapping factor $F_o$, which reflects how many tasks have selected a particular weight. The merged update for the weights is:

$$\Delta W_m = \frac{1}{F_o} \sum_{i=1}^{N} \Delta W_i, \tag{4}$$

where $F_o = \min(\sum_{i=1}^{N} M_i, \mathbf{1})$, is the element-wise sum of the masks $M_i$ and $\sum_{i=1}^{N} M_i$ represents the count of how many tasks selected each weight element. We capped at 1 to prevent division by zero. Finally, the merged model weights are obtained by adding the merged sparse update $\Delta W_m$ to the base model weights: $W = W + \Delta W_m$.

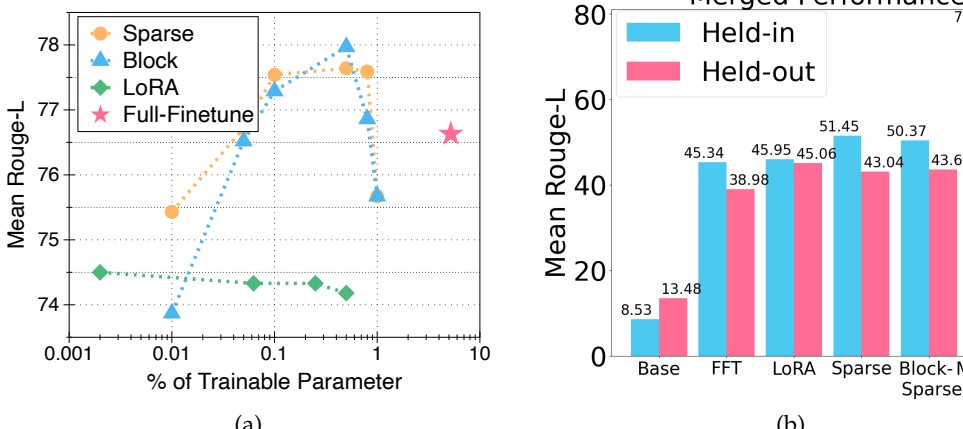

Figure 3: *(a)* Comparison of sparse adapter, LoRA, and Full FT on single task performance. We report the average rouge-L performance across 20 tasks, showing performance variations with different trainable parameter ranges by adjusting the LoRA tuning rank and the fraction of parameters for the sparse adapter in log scale. *(b)* Merged performance on both held-in (20 tasks) and held-out tasks (10 tasks).

## 4 Experiments

We begin by evaluating the performance of our sparse adapter in comparison to LoRA and full fine-tuning in Section 4.1. Then, we investigate alternative merging techniques and examine various parameter selection methods applied to sparse adapters in Section 4.2. In Section 4.3, we explore the effects of weight interference in the sparse adapters, the scalability of merging methods with an increasing number of experts, and potential approaches for further parameter reduction.

**Setup** We conduct our experiments using the FLAN dataset (Longpre et al., 2023) and sample 20 held-in and 10 held-out tasks following (Ostapenko et al., 2024), where each task is sub-sampled to 10,000 examples. Within these samples, 1,000 are allocated for validation and early stopping. For parameter-efficient fine-tuning, we load the base model $W$ in bfloat16 format and trainable parameters $\hat{W}$ in float32. As the base model, we use Phi-3-mini-4k-instruct (3.8B parameters) (Abdin et al., 2024) for all our finetuning experiments. For each FLAN task, we finetune the model for 5 epochs. For hyperparameter sweeps, we randomly select 5 tasks from the 20 held-in tasks and keep them fixed. For sparse adapter, we conduct multiple hyperparameter searches on layer selection, learning rate, and block-size for block-sparsity, with details provided in the Appendix A.

### 4.1 Single Task Finetuning

For single-task performance comparison, we finetune 20 FLAN tasks for 5 epochs and provide the mean Rouge-L performance. In Fig. 3(a), we compare the performance of sparse adapter across different sparsity levels, alongside LoRA (Hu et al., 2021b) and full fine-tuning (FFT). Specifically, we adjust the sparsity by varying the parameter $kr$, which represents the percentage of parameters retained during training. For instance, $kr = 0.01$ corresponds to a sparsity of 99%. In our experiments, both sparse adapter and LoRA train the QKV layers. To ensure a fair comparison, we also adjust the rank of LoRA accordingly, maintaining consistency across the evaluation.

Our results in Fig. 3(a) show that our sparse adapter with $kr = 0.01$ (99% sparsity) outperforms the fully fine-tuned model. Performance improves as $kr$ increases up to 0.5, after which we observe a gradual decline in performance at $kr = 0.8$. Interestingly, when $kr = 1$,

| Method | Merging | % param trainable | Hparam | Mean Rouge-L Held-In | Held-Out |
|---|---|---|---|---|---|
| **Full-Finetune** | Averaging | 100% | | 45.33 | 38.98 |
| | Task-Arithmetic | 100% | ✔ | 39.40 | 25.31 |
| | Ties | 100% | ✔ | 50.21 | **46.16** |
| | Breadcrumbs | 100% | ✔ | 39.44 | 25.33 |
| **LoRA** | Averaging | 1.54% | | 45.96 | 44.01 |
| **Element-Sparse** | Averaging | 2.37% | | **51.44** | 43.09 |
| **Block-Sparse** | Averaging | 2.37% | | 50.37 | 43.61 |
| **Multitask** | - | 100% | | 77.15 | 54.00 |

Table 1: Comparison of merging performance between full fine-tuning and PEFT methods across various merging techniques. We report the Rouge-L score for all approaches. The percentage of training parameters for each expert is presented, comparing LoRA ($r = 128$) and sparse adapters ($kr = 0.1$).

which corresponds to dense training of the QKV layers, performance drops further. This suggests that training the QKV layers alone, even when fully dense, is not sufficient for optimal performance. Instead, selecting an appropriate subspace of parameters is crucial to achieving better results.

## 4.2 Model Merging Performance

We explore the merging of 20 expert models and compare their performance to multitask training. Fine-tuning a base model on specialized tasks can often result in a loss of generalization ability (Wortsman et al., 2022). As a result, we investigate whether combining the expertise of multiple sparse adapters can improve performance on held-out data over the multitask performance. We select 10 tasks from the FLAN dataset to evaluate the held-out performance of the models. We found that the sparse adapter performs best on both held-in and held-out tasks when $kr = 0.1$. Therefore, we use $kr = 0.1$ for the performance comparison in the model merging experiments. Result for varying $kr$ is presented in Fig. 11 and discussed in Appendix B.

By default, we use simple uniform weight averaging for merging (Wortsman et al., 2022). We benchmark alternative merging methods in the following section. For sparse merging, we adopt our weighted averaging accounting for the overlap between sparse models (Eq. (4)). In Fig 3(b), we compare the merging performance of FFT and LoRA as baseline models. In the case of FFT, we average the weights of multiple models equally. For LoRA, we compute the average over the low-rank adapters after the outer product.

**Performance Compared to Multitask Training** Multitask training outperforms all merged methods on both held-in and held-out tasks. The sparse adapter demonstrates the best model merging performance for held-in tasks and performs competitively with LoRA on held-out tasks. Unlike what we have seen in recent work, (Yadav et al., 2024b; Aakanksha et al., 2024) we observe that multitask training outperforms model merging in terms of held-out generalization when scaling to 20 expert datasets. We hypothesize that training on 20 expert datasets with natural language prompts across diverse domains enables the multitask model to generalize more effectively across various types of instructions. This allows multitask training to capture broader patterns that individual expert models may overlook, making it more effective for held-out tasks.

**Performance Compared to Different Merging Methods** Various model merging methods typically fine-tune a pre-trained base model and compute a *task vector* (Ilharco et al., 2023) by subtracting the original model weights from those after fine-tuning on a specific task: $\tau^n = W^n_{\text{finetune}} - W$. These task vectors $\{\tau\}^N_{n=1}$ are then used to adjust the behavior of the merged model. One straightforward approach, *Task-Arithmetic* (Ilharco et al., 2023), sums the task vectors and computes a weighted merge with the base model: $W_{\text{new}} = W + \lambda \sum^N_{n=1} \tau^n$. To address parameter interference caused by different task vectors, methods such as *TIES* (Yadav et al., 2023) trim less impactful task vectors by setting them to zero, resolving sign

conflicts through majority voting among the vectors. *Breadcrumbs* (Davar, 2024) proposes filtering out outliers and removing negligible perturbations from the task vectors to improve merging performance. We also compare *Uniform weight-averaging* (Wortsman et al., 2022), which involves averaging the weights of multiple models fine-tuned on different tasks uniformly. We leave out computationally expensive approaches, such as those involving Fisher matrices (Matena & Raffel, 2022b), backward passes (Yang et al., 2024a), or computing model activations (Jin et al., 2023a), as these methods do not scale well with large models or a high number of expert. As shown in Tab. 1, despite employing simple weight averaging, sparse adapters achieve the highest Rouge-L scores (sparse: 51.44, block-sparse: 50.37) compared to other merging methods across 20 held-in tasks. Although sparse adapters utilize only 2.37% of the trainable parameters in comparison to FFT, they surpass the FFT-averaging by 13.48% and 11.12% in performance.

**Compare Different Parameter Selection Criteria** We compare performance across various methods, which serve as criteria for parameter selection to identify task-specific masks $M$. Unlike previous sparse training methods (Ansell et al., 2022; Panda et al., 2024) that involve a full fine-tuning stage, we apply parameter selection methods directly within the framework outlined in Algorithm 1. *Connection-Sensitivity* (CS): The original implementation of Lee

| Method | Individual | Merged | |
| --- | --- | --- | --- |
| | | Held-In | Held-Out |
| **MCS** (Ours) | **77.54** | **51.44** | 43.03 |
| **CS** | 75.92 | 49.49 | 43.16 |
| **GM** | 76.64 | 48.04 | 42.89 |
| **WM** | 77.28 | 47.37 | 36.54 |
| **GD** | 76.23 | 46.23 | **43.38** |

Table 2: Comparison of performance across different scoring methods as Individual experts and Merged model. We provide the mean RougeL score evaluated over the held-in and held-out tasks.

et al. (2018) utilizes the absolute over the scoring function $|\Delta W \frac{\partial \mathcal{L}}{\partial \Delta W}|$. *Grow and Drop* (GD): Ansell et al. (2024) employs two distinct functions: the weight difference from initialization $|\Delta W - \Delta W_0|$ for determining drops, and the gradient magnitude $|\frac{\partial \mathcal{L}}{\partial \Delta W}|$ for deciding which parameters to grow. *Gradient Magnitude* (GM): Following (Frankle & Carbin, 2019; Ansell et al., 2022; Panda et al., 2024), we select parameters based on the gradient magnitude $|\frac{\partial \mathcal{L}}{\partial \Delta W}|$. *Weight Magnitude* (WM): Additionally, we use a simple parameter selection method based on the weight magnitude $|\Delta W|$. We compare the performance of these different methods in Table 2. While the held-out performance is competitive, we see a clear advantage of using Max Connection sensitivity (Equation 2).

### 4.3 Additional Analyses

**Impact of Merging with Increasing Number of Experts** We study the impact of model-merging as we increase the number of merged experts. In Fig. 6, we evaluate the performance on various merging methods as the number of merging experts, denoted as $N$, increases from 2 to 20, with the values $N = \{2, 5, 10, 20\}$. For each value of $N$, we sample sparse adapters, merge them and test performance on the test sets of the corresponding tasks. We conduct 10 trials and compute the mean performance across these trials. The figure presents both the performance variation and the mean performance for each $N$. Results demonstrate that merging sparse adapters offers clear advantages across different numbers of experts, surpassing alternative merging techniques.

**Impact of Weight Interference During Merging** We investigate the impact of weight interference during model merging for sparse adapters. When merging sparse adapters, we can identify two sources of interference that can be detrimental for a task performance: interference occurring by changing the values of the parameters for a given task ("masked"), and interference that occur by changing the values of the parameters not selected by a given task ("non-masked"). We present a simple visual representation of model merging of two sparse adapters in Fig. 4 along with "masked + non-masked" interference and "masked" only interference.

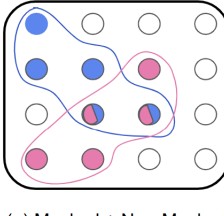 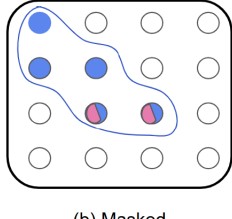 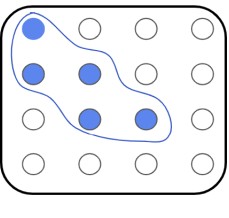

(a) Masked + Non-Masked      (b) Masked      (c) No Interference

Figure 4: Visual representation of sources of weight interference for task $A$ (blue) due to parameters of another task $B$ (pink) for sparse adapter uniform weight averaging. (a) Task $A$ performance is affected by changes of values of parameters in $M_A$ and outside of $M_A$. (b) We isolate how much of the interference is due to changes in masked region.

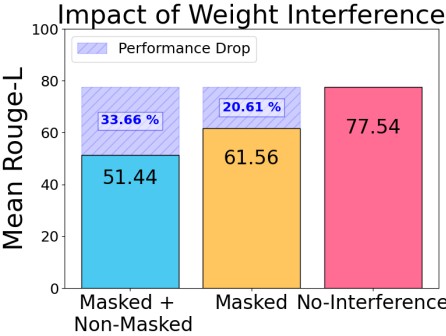

Figure 5: Impact of weight interference due to merging sparse adapters. We compare the mean Rouge-L score of the 20 held-in tasks.

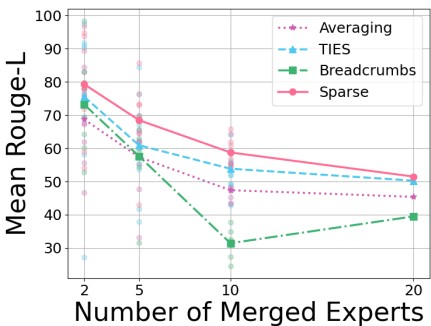

Figure 6: Performance comparison of different merging methods varying the number of merged experts and evaluated on the held-in tasks with mean computer over 10 trials.

To quantify the effect of interference in the masked weight space, we evaluate the merged weights using the task mask: $\Delta W_m^* = m_i * \Delta W_m$. As illustrated in Fig. 5, interference in the masked space results in a 20.61% performance degradation relative to single-task performance, which serves as the "no-weight interference" baseline. The overall performance drop of the merged model reflects the combined effects of interference in both masked and non-masked regions. As shown in Fig. 5, we observe an additional 13.05% performance reduction, attributable to interference in the non-masked weight space. We conclude that the degradation in held-in performance is primarily driven by weight interference due to overlap within the masked region, rather than parameter changes occurring outside the sparse masks during merging. This insight motivates future work on designing sparse adapter training mechanisms that can (a) reduce weight interference and (b) ensure that training is invariant to parameter modifications outside the masked weights.

**Performance Comparison with Active Layer Reduction** In addition to layer-wise sparsity, we introduce a mechanism to reduce the number of active layers during sparse adapter training. This mechanism identifies and removes less important layers based on their significance to the model. For a given sparsity level, we rank all the parameters of the model according to their importance scores using Eq. (2). The threshold for importance is defined by the $k^{th}$ highest score among all the model parameters, which acts as a cutoff. Any layer whose parameters fall below this threshold is considered less important and is dropped from the training process. As a result, only the layers with the most important parameters

| Method | Held-In | Held-Out | Number of active layers |
|---|---|---|---|
| **Sparse-Adapter** | **51.44** | **43.03** | 32 |
| **Sparse-Adapter + Layer-drop** | 48.45 | 41.29 | $20.02 \pm 2.08$ ( **37.5%** $\downarrow$) |

Table 3: Performance of the sparse adapter with active layer drop, evaluated on both held-in (20 tasks) and held-out (10 tasks) tasks.

are kept active, thus reducing the number of layers involved in sparse-adapter training:

$$\text{Threshold} = \text{Quantile}_k \left( \bigcup_{l=1}^{L} \mathcal{S}_i(\hat{W}; D_{\mathcal{T}}) \right), \tag{5}$$

$$\text{drop}(l) = \begin{cases} 1 & \text{if } \mathcal{S}(\hat{W}_l; D_{\mathcal{T}}) < \text{Threshold}, \quad l = 1 \text{ to } L \\ 0 & \text{if } \mathcal{S}(\hat{W}_l; D_{\mathcal{T}}) \geq \text{Threshold}, \quad l = 1 \text{ to } L \end{cases} \tag{6}$$

In the above equations, $\mathcal{S}(\hat{W}_l; D_{\mathcal{T}})$ represents the importance score of the parameters in layer $l$, and Quantile$_k$ calculates the $k^{th}$ highest importance score across all layers. Layers with scores below the threshold are dropped, while those with scores above or equal to the threshold are retained. The results are summarized in Tab. 3. The "Sparse Adapter with Layer-Drop", which reduces the number of active layers to $\approx$20 (a 37.5% reduction, mean layer reduction computed for 20 tasks), leads to a slight drop in performance: 48.45 for held-in and 41.29 for held-out. Despite 37% layer reduction sparse adapter outperform full-finetuning model merging methods for held-in tasks (presented in Table 1). While this reduction results in a modest decrease in performance, it demonstrates the trade-off between reducing the number of active layers and accuracy. These results highlight the potential for balancing model efficiency and performance by controlling the number of active layers. By utilizing a global selection criterion to prune low-priority layers, we can optimize the model for resource-constrained environments, sacrificing a slight performance efficiency.

## 5 Conclusion

In this paper, we explore the potential of sparse adapters as efficient building blocks for modular architectures in multitask learning. Our training approach offers conceptual simplicity while outperforming LoRA and full fine-tuning across 20 tasks. Additionally, our model merging experiments demonstrate that sparse adapters maintain strong performance on held-in tasks and competitive held-out generalization. Unlike full-finetuning methods that degrade when scaled to 20 experts, sparse adapters remain effective and consistently outperform conventional model merging techniques. Despite these advantages, a performance gap remains when compared to multitask training. Ablation studies indicate that the degradation in held-in performance is primarily caused by weight interference resulting from parameter modifications within the sparse mask, rather than changes to parameters outside the sparse mask during merging. This finding highlights a key insight that motivates future work on designing more effective sparse adapters. Our study highlights the potential of sparse adapters as a scalable and efficient solution for constructing modular architectures, particularly as the number of tasks increases. These findings open avenues for future research aimed at closing the gap and further improving model merging performance.

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

# A    Hyperparameter Search

**Which Layer Should We Sparsify?**    We sparsify only the $QKV$ parameters in the attention layers of each transformer module. While the output projection layer, $O$, can also be fine-tuned (Hu et al., 2021b), we observe that fine-tuning just the $QKV$ parameters results in better performance. An alternative approach is to fine-tune the MLP layers. To investigate this, we compare the fine-tuning of the 'down-proj' $MLP$ layer of Phi-3. Empirical results are shown in Fig. 7, where we compare performance across both single-task and merged models, evaluating on both held-in and held-out datasets after fine-tuning with a keep-ratio of 0.1. While fine-tuning the $MLP$ layers improves held-in model merging performance, it negatively impacts held-out performance, leading us to prefer fine-tuning the $QKV$ layers.

**Learning Rate Hyperparameter Tuning:**    We conduct a hyperparameter sweep to identify the optimal learning rate for fine-tuning the sparse-adapter, LoRA and FFT model. The mean performance presented in Fig. 9 is evaluated across five fixed FLAN tasks, with learning rates varied at $1e^{-3}$, $1e^{-4}$, and $1e^{-6}$ to assess their impact on model performance.

**Tuning Block-Size Hyperparameter for Block-Sparse:**    We conduct an exploration of different block sizes, $B$ in block-sparse training to identify the optimal setting. As shown in Fig. 10, we compare the performance of block-sparse training (with kr=0.1) across block sizes of 8, 16, and 32. Our results reveal that a block size of 16 delivers the best overall Rouge-L score for five individual tasks.

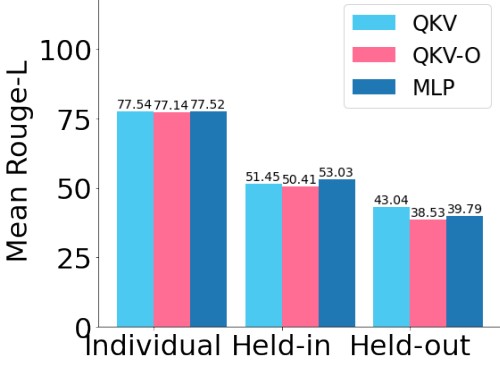
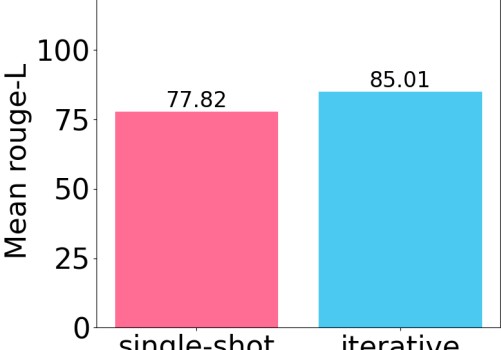

Figure 7: Performance of Sparse-Adapter ($kr = 0.1$) on training $QKV$, $QKV$-$O$, $MLP$ layers in Phi-3. Mean Rouge-L computed across 20 individual tasks and merged Performance for 20 held-in and 10 held-out tasks.

Figure 8: Performance of Sparse-Adapter ($kr$=0.1) on single-shot vs iterative update of the subspace. We compare the mean rouge-L performance of 5 individually trained tasks.

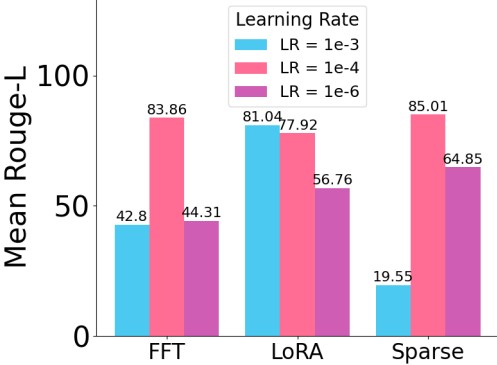

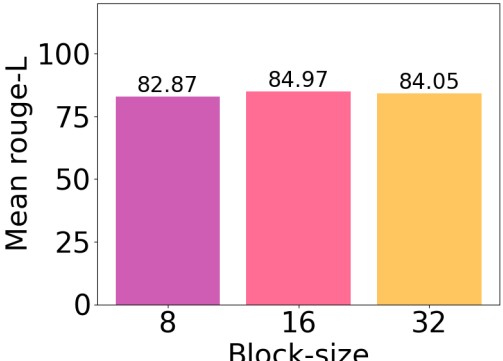

Figure 9: Performance of different methods under varying learning-rate. We compare the mean Rouge-L performance of 5 individually trained tasks to decide the best learning rate for each method.

Figure 10: Performance of Block-Sparse-Adapter (*Kr*=0.1) on varying different block-size. We compare the mean Rouge-L performance of 5 individually trained tasks to decide the block-size.

## B   Additional Results

**Performance under Varying Sparsity:** Fig. 11 shows the performance improvement of the sparse adapter method over the Phi-3 base model at different *kr* values. We find that $kr = 0.1$ provides the best merging performance for both held-in and held-out datasets. Although the best single-task performance without merging is achieved at a $kr = 0.5$ (Fig. 3), the increased weight population leads to greater interaction between weights, causing weight corruption that negatively impacts merging performance. This observation demonstrates a parameter saturation effect: as the number of parameters increases, the learning complexity of sparse training grows, leading to improved merging performance up to $kr = 0.1$. However, beyond this point, more weight conflicts arise, leading to performance degradation when *kr* exceeds 0.1.

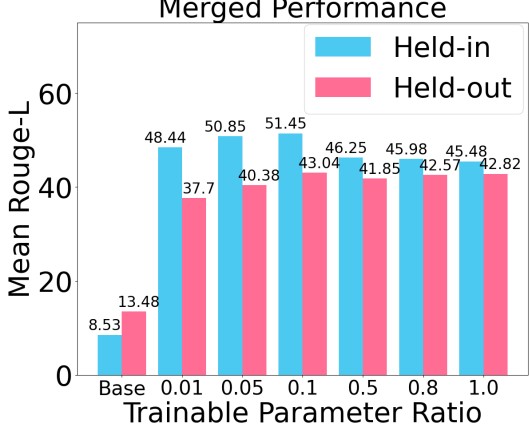

Figure 11: Sparse adapter performance comparison across different values of *kr* on held-in (20 tasks) and held-out (10 tasks) after merging.

**Can we recycle weights for multitask training?** After sparse model merging we explore whether we can recycle the weight for further in multitask training. Our results show a significant performance improvement following the initial merging, with further gains achieved through multitask training shown in Table 4. After 4 epochs of sparse single-task training, we conduct additional $N = \{1, 5\}$ epochs of multitask training on the merged model. In the second stage of fine-tuning, we compare the performance of two approaches: (a) fine-tuning only the merged sparse weights, $\Delta W_m$, and (b) making the entire model $W$ trainable, where $W \leftarrow W + \Delta W_m$.

## C   Additional Discussion

**Computational Considerations of Model Mering Methods:** As shown in Table 1, merging methods such as Task-Arithmetic (Ilharco et al., 2023), TIES (Yadav et al., 2023), and Breadcrumbs Davar (2024) require hyperparameter tuning to achieve optimal performance.

| Training Configuration | Held-In | Held-Out |
|---|---|---|
| **Sparse-Merge + ST (1 epoch)** | 69.70 | 50.96 |
| **Sparse-Merge + ST (5 epoch)** | 76.45 | - |
| **Sparse-Merge + FFT (1 epoch)** | 73.90 | 43.04 |
| **Sparse-Merge + FFT (5 epoch)** | 76.72 | - |
| **Multitask** | 77.15 | 54.00 |

Table 4: Comparison of training performance across different configurations: 5 epochs for MT training and sparse merging with FFT and Sparse Training (ST) after 4 epochs of parallel training and N = {1,5} epoch of multitask training after merging.

We used the recommended hyperparameters for these methods. While sparse adapter only involves averaging weights when merging models, both TIES and Breadcrumbs require a TopK operation for each expert model to filter parameters, which is computationally expensive. The time complexity of the TopK operation is typically $O(n \log k)$, where $n$ is the number of elements in the input tensor, and $k$ is the number of top elements to retrieve. As the number of model parameters increases, the computational cost of this operation grows significantly.

**Memory Usage**    In sparse training, memory management revolves around two key stages: (1) mask selection and (2) parameter update. During mask selection, the process require $\Delta W$ and a mask $M$ which are the same dimension of $QKV$ layer. When updating parameters, we can store only the trainable weights selected using the scoring function and corresponding index locations in GPU while we store the unmasked weight matrix in CPU for next mask calculation step. This strategy optimizes memory usage during training. After completing the first epoch, $M$ is fixed, and we can discard the unmasked weights.

