# OpenReview forum: "Exploring Sparse Adapters for Scalable Merging of Parameter Efficient Experts"
_colmweb.org/COLM/2025/Conference — COLM 2025_

### Official Review · Reviewer_u9fH · 2025-05-11

**Rating:** 6
**Confidence:** 3
**Ethics Flag:** 1

**Summary:**

The paper proposes to use sparse adapters for merging to achieve better performance in downstream tasks in LLM evaluation. The authors use dynamic sparse masks, and a new criterion to select trainable parameters. For model merging the authors choose to use a simple averaging algorithm with weight masks as coefficients. The authors conduct evaluation on a series of tasks from the FLAN collections and show that their algorithm can achieve better results.

**Reasons To Accept:**

- The algorithm is easy to implement, and can achieve good performance on multiple tasks.
- The paper is well written when describing the algorithm, making it easy to reproduce.

**Reasons To Reject:**

- In my opinion the '% of trainable parameters' is misleading. The authors keep choosing a different set of hyperparameters to update, but always have the full parameters in the model. In my opinion in this case the trainable parameters should be considered as the percentage of parameters that has been changed during the training procedure.
- If the above statement is correct, then the comparison with baseline methods need to be revisited, especially with Task-Arithmetic and Ties.
- What is the practical usage of block-wise sparsity here? In sparsity literature, block wise sparsity is mainly for acceleration. However, the acceleration cannot exist here since the authors keep the full weight matrices; the performance is not higher either compared to using the unstructured version. Could the authors explain how is the block-wise sparsity is actually useful in the context of this paper?
-

---

> ### Author Response · Authors · 2025-06-02
> **Rebuttal by Authors**
>
> We appreciate the reviewer’s positive feedback on the clarity and reproducibility of our method. We appreciate the opportunity to clarify. While we store the full parameter matrix during the first epoch for mask selection, only a fixed sparse subset (e.g., 5%) is active and updated throughout training and inference. After the first epoch, the mask is fixed and the rest of the weights are discarded. As the final model retains and modifies only this subset, we believe the reported percentage of trainable parameters fairly reflects the effective optimization budget. In practice, when sharing and merging models, it is convenient for these adapters to be lightweight, a property which our method satisfies. We will clarify this further in the revised version.
>
> We agree that block-wise sparsity is traditionally used for hardware acceleration. In this work, our focus is on exploring how far block-sparse methods can be improved in the context of model merging. While model merging enables decentralized training and composability, our long-term goal is to extend these techniques to the merging of Mixture-of-Experts (MoE) models, where block-wise sparsity can be leveraged for efficient hardware acceleration. This work represents an initial step in that direction.

---

> > ### Comment · Reviewer_u9fH · 2025-06-08
> >
> > Thank you for your clarification - please do involve the above discussion as it helps reduce a lot of misunderstanding.

---

> > > ### Author Response · Authors · 2025-06-08
> > > **Response by authors**
> > >
> > > Thank you for your comment. We will be sure to include this clarification in the final version of the paper.

---

### Official Review · Reviewer_GEzS · 2025-05-13

**Rating:** 7
**Confidence:** 3
**Ethics Flag:** 1

**Summary:**

This paper studies the use of sparse adapters as an alternative to LoRA for parameter-efficient fine-tuning and for merging multiple task-specific experts into a single model. The authors propose a simple method for training these sparse adapters, which involves identifying a subset of weights using a saliency-based score and updating this mask dynamically only during the first epoch of training.

The core contributions are:
- Training methodology for sparse adapters that outperforms both LoRA and full fine-tuning for single-task performance, even at high sparsity levels (99%).
- An extensive investigation into merging up to 20 such sparse adapters. The results show that merged sparse adapters achieve superior in-distribution performance compared to merged LoRA experts or merged fully fine-tuned models.
- An analysis of different parameter selection criteria, the impact of the number of merged experts, sources of weight interference, and a method for active layer reduction.
- While the proposed sparse adapters show strong held-in performance post-merging, the paper acknowledges that achieving strong held-out generalization remains a challenge for all merging methods considered, though sparse adapters are competitive.

**Reasons To Accept:**

- Novel and Effective Sparse Adapter Training: The proposed method for training sparse adapters i.e. dynamic mask selection using MCS is conceptually simple and empirically effective, outperforming strong baselines like LoRA and full fine-tuning in single-task settings.
- Scalable Merging Experiments: The paper pushes the boundary of merging experiments by considering up to 20 task experts. This is a more realistic and challenging scenario than typically studied (often 2-3 tasks) and provides valuable insights into the scalability of different merging approaches.
- Superior In-Distribution Merging Performance: The key finding that merged sparse adapters outperform merged LoRA and full fine-tuning on in-distribution tasks is significant. This suggests less destructive interference when combining sparse adapters compared to other methods.
- Comprehensive Ablation Studies:
    - Comparison of different parameter selection criteria justifying their MCS choice.
    - Impact of increasing the number of merged experts.
    - Investigation into weight interference, distinguishing between masked and non-masked regions.
    - Exploration of active layer reduction.

**Reasons To Reject:**

- Held-Out Generalization Gap: While the paper is upfront about this, the held-out performance of merged sparse adapters, while competitive, does not clearly surpass merged LoRA (Table 1: Sparse 43.03/43.61 vs LoRA 44.01) and significantly lags behind multitask training (54.00). Since improved generalization is a primary motivation for model merging, this remains a notable limitation
- Computational Cost of Mask Identification: The MCS-based mask selection requires a forward-backward pass every 100 steps during the first epoch. While this might be less than a full fine-tuning epoch required by some other methods, it's an overhead not present in LoRA. A more explicit discussion or comparison of this training-time overhead would be beneficial.

---

> ### Author Response · Authors · 2025-06-02
> **Rebuttal by Authors**
>
> We thank the reviewer for a thorough review. We appreciate the reviewer’s observation and agree that generalization to held-out tasks remains a significant challenge. While multitask training often struggles in this regard, model merging has emerged as a practical alternative, particularly in settings where multitask training is infeasible. It enables decentralized training of specialized experts and allows for greater scalability and modularity. That said, to further enhance generalization on held-out data, we believe it is important to also incorporate insights from the perspective of data distribution, which may complement the strengths of model merging.
>
> During mask selection, we use the same batch size as in training, and performing this once every 100 steps increases the training overhead by only about 1%, which we consider negligible.

---

### Official Review · Reviewer_MeHe · 2025-05-13

**Rating:** 6
**Confidence:** 4
**Ethics Flag:** 1

**Summary:**

The paper studies the benefit of training sparse adapters for individual tasks and then selectively merging them for multi-task learning. Due to the sparseness of the weight matrices, the resulting combined matrix for the multi-task setting has a minimal number of cooccurring weights, and hence (arguably) a reduced interference. The proposed merging method is the aggregation of delta weights on each weight position.

The authors conduct a set of experiments on the FLAN dataset, and evaluate the performance in terms of held-in and held-out tasks, and compare the method with "basic" baseline LoRA method as well as full fine-tuning.

**Questions To Authors:**

I believe, this paper is highly relevant to the work as it provides a comparative analysis of various combinatorial methods:

**Reasons To Accept:**

- The method is sensible and provide a good path forward for effective weight merging in multi-task settings
- The provided improvement could have practical benefits.

**Reasons To Reject:**

- The experiments only contains a basic set of baselines. It is not clear how well the method works under different variations. The paper below e.g., studies the effect of various ways of merging/combining.

Holtermann, C., Frohmann, M., Rekabsaz, N., & Lauscher, A. (2024, March). What the Weight?! A Unified Framework for Zero-Shot Knowledge Composition. In Findings of the Association for Computational Linguistics: EACL 2024 (pp. 1138-1157).

---

> ### Author Response · Authors · 2025-06-02
> **Rebuttal by Authors**
>
> We thank the reviewer for pointing out this work. We will cite it in the revised manuscript and clarify the distinction in the literature review. While the suggested paper focuses on composing knowledge modules for zero-shot generalization through optimal domain composition, our work focuses on studying the specific properties of sparsity in the context of model merging.

---

### Comment · Area_Chair_UPDa · 2025-06-06
**Brief Clarification on Method and Contributions**

Dear Reviewers,

  Thank you for your positive reviews. The authors have provided brief clarifications:

  - Clarified that only the sparse subset (5%) is retained after training
  - Acknowledged held-out performance remains challenging for all merging methods
  - Positioned their work as complementary to the suggested reference

  Given the already positive reception (6, 7, 6), this serves mainly as confirmation of your assessments.

---

### Decision · Program_Chairs · 2025-07-08

**Decision:**

Accept

**Comment:**

All reviewers liked it (6,7,6). Key win: sparse adapters beat LoRA for merging models, with very robust experimental scale (20 tasks vs typical 2-3). The method is simple, adds minimal overhead (~1%), and while held-out performance is not perfect, that is true for all merging methods. Authors clarified that only 5% sparse weights are kept, addressing the main concern. Straightforward accept.